# HEROFILTER: Adaptive Spectral Graph Filter for Varying Heterophilic Relations

**Shuaicheng Zhang**[*]
Virginia Tech
zshuai8@vt.edu

**Haohui Wang**[*]
Virginia Tech
haohuiw@vt.edu

**Junhong Lin**
MIT
junhong@mit.edu

**Xiaojie Guo**
IBM Research
xiaojie.guo@ibm.com

**Yada Zhu**
IBM Research
yzhu@ibm.com

**Si Zhang**
Meta AI
sizhang@meta.com

**Dongqi Fu**
Meta AI
dongqifu@meta.com

**Dawei Zhou**
Virginia Tech
zhoud@vt.edu

## Abstract

Graph heterophily, where connected nodes have different labels, has attracted significant interest recently. Most existing works adopt a simplified approach - using low-pass filters for homophilic graphs and high-pass filters for heterophilic graphs. However, we discover that the relationship between graph heterophily and spectral filters is more complex - the optimal filter response varies across frequency components and does not follow a strict monotonic correlation with heterophily degree. This finding challenges conventional fixed filter designs and suggests the need for adaptive filtering to preserve expressiveness in graph embeddings. Formally, natural questions arise: *Given a heterophilic graph $\mathcal{G}$, how and to what extent will the varying heterophily degree of $\mathcal{G}$ affect the performance of GNNs? How can we design adaptive filters to fit those varying heterophilic connections?* Our theoretical analysis reveals that the average frequency response of GNNs and graph heterophily degree do not follow a strict monotonic correlation, necessitating adaptive graph filters to guarantee good generalization performance. Hence, we propose **HEROFILTER**, a simple yet powerful GNN, which extracts information across the heterophily spectrum and combines salient representations through adaptive mixing. HEROFILTER's superior performance achieves up to **9.2%** accuracy improvement over leading baselines across homophilic and heterophilic graphs.

## 1 Introduction

Graph Neural Networks (GNNs) have emerged as powerful tools for learning from graph-structured data across a broad range of domains [1, 2, 3, 4, 5, 6, 7, 8, 9]. Despite their empirical success, GNNs are known to suffer performance degradation when applied to graphs that exhibit *heterophily*, a structural property where connected nodes tend to have dissimilar features or labels. This stands in contrast to the *homophily* assumption, foundational to many GNN architectures, which presumes that adjacent nodes share similar attributes. A growing body of work [10, 11, 12, 13, 14] has shown that this mismatch in assumptions significantly limits the effectiveness of conventional GNNs, often rendering them less effective than simple multilayer perceptrons (MLPs).

To better understand and mitigate the limitations of GNNs under heterophily, researchers have adopted a spectral view grounded in graph signal processing (GSP) [15, 16, 17, 18]. In this framework, graph signals are decomposed into frequency components via the eigen decomposition of the graph Laplacian, where low-frequency components capture smooth variations across the graph, and high-

---

[*]Equal contribution.

frequency components capture rapid, localized changes. Classic GNNs such as Graph Convolutional Network (GCN) [19] and Graph Attention Network (GAT) [20] can thus be interpreted as applying low-pass filters, effectively amplifying low-frequency components and suppressing high-frequency noise [15]. This design aligns well with homophilic structures, where relevant information is concentrated in the low-frequency spectrum. However, it is often inadequate for heterophilic graphs, where informative signals may lie in higher-frequency bands.

To address this, recent works have proposed filter-based GNNs that explicitly incorporate high-frequency information [22, 23], and empirical studies have supported the effectiveness of high-pass or mixed filters for heterophilic settings [24, 23]. Nevertheless, most existing approaches rely on a simplifying assumption: low-pass filters are used for homophilic graphs, and high-pass filters are used for heterophilic ones. This binary perspective assumes a monotonic relationship between the heterophily level of a graph and its optimal spectral filter, an assumption that we show is often violated in practice.

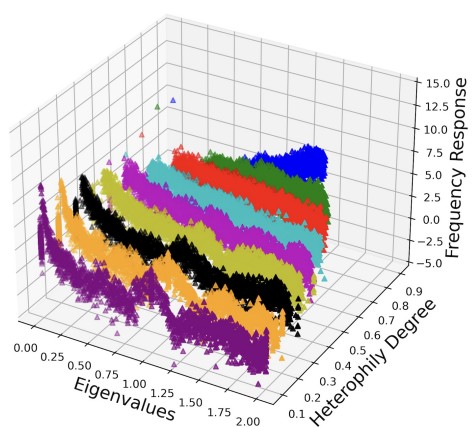

Figure 1: 3D visualization showing the underlying relationship among eigenvalues, heterophily degree, and frequency response (in log scale). In particular, we synthesized nine graphs with heterophily degrees ranging from 0.1 to 0.9. Each curve with a unique color corresponds to a well-trained spectral graph filter [21] on each graph.

To investigate this assumption, we conduct controlled experiments on synthetic graphs with varying heterophily degrees. As shown in Figure 1, we observe that the frequency responses of trained spectral filters [21] exhibit complex, non-monotonic behavior across the spectrum. For example, in graphs with low heterophily (e.g., 0.1), the learned filters reveal significant activation in both low- and mid-frequency bands, contrary to the expectation of a purely low-pass response. Similarly, graphs with high heterophily (e.g., 0.9) do not exhibit a purely high-pass response, retaining strong low-frequency components. These findings suggest that the relationship between heterophily and spectral response is more intricate than previously assumed, and that fixed filter types (low-pass or high-pass) are insufficient for capturing this complexity. These observations motivate two central research questions:

**Q1 (Theoretical Understanding):** *How can we characterize the relationship between graph heterophily, spectral filters, and the prediction performance of GNNs?*

**Q2 (Adaptive Computation):** *How can we design adaptive filters and GNN models that perform robustly across graphs with varying and possibly mixed heterophily patterns?*

To address Q1, we conduct a theoretical analysis that formally establishes the connection between the heterophily degree of a graph, its spectral representation, and the performance of GNNs. We introduce a novel spectral-domain measure of heterophily and derive an error bound that highlights the limitations of fixed filtering strategies and motivates the need for adaptivity in spectral design.

To address Q2, we propose **HEROFILTER**, a novel GNN architecture inspired by the MLP-Mixer [25, 26]. HEROFILTER consists of two key components: (1) a **Patcher** that identifies spectrally relevant neighbors for each node via adaptive polynomial filters, and (2) a **Mixer** that aggregates and transforms patch representations along both the patch and feature dimensions. This architecture allows HEROFILTER to modulate its spectral response based on the graph structure and heterophily level, while remaining computationally efficient. We further introduce **Fast-HEROFILTER**, a scalable variant that avoids eigen decomposition through efficient approximation.

Beyond conceptual proof and proposed theories, we conduct extensive experiments on homophilic, heterophilic, and large-scale real-world graphs. HEROFILTER consistently achieves state-of-the-art accuracy, improving over strong baselines by up to 9.2%. Data and code are available [2].

---

[2] https://github.com/zshuai8/HeroFilter

## 2 Preliminaries

**Notation.** We denote scalars by regular lowercase letters (e.g., $c$), vectors by bold lowercase letters (e.g., $\mathbf{r}$), and matrices by bold uppercase letters (e.g., $\mathbf{X}$). A graph is represented as $\mathcal{G} = (\mathcal{V}, \mathcal{E}, \mathbf{X})$, where $\mathcal{V}$ is the set of nodes, $\mathcal{E} \subseteq \mathcal{V} \times \mathcal{V}$ is the set of edges, and $\mathbf{X}$ is the node feature matrix. Let $n = |\mathcal{V}|$ be the number of nodes. We denote by $\mathbf{A}$ and $\mathbf{L}$ the adjacency and Laplacian matrices, respectively, and by $\tilde{\mathbf{A}}$ and $\tilde{\mathbf{L}}$ their normalized forms. A summary of key notations is provided in the Appendix B.

**Graph Spectral Filters.** Spectral GNNs leverage the graph signal processing framework to define convolutional operations in the spectral domain. The eigendecomposition of the normalized Laplacian $\tilde{\mathbf{L}}$ yields $\tilde{\mathbf{L}} = \mathbf{U}\mathbf{\Lambda}\mathbf{U}^\top$, where $\mathbf{U} \in \mathbb{R}^{n \times n}$ is the matrix of orthonormal eigenvectors forming the graph Fourier basis, and $\mathbf{\Lambda}$ is a diagonal matrix containing the corresponding eigenvalues. Given a graph signal $\mathbf{x} \in \mathbb{R}^n$, its graph Fourier transform is defined as $\hat{\mathbf{x}} = \mathbf{U}^\top \mathbf{x}$, and the inverse transform is $\mathbf{x} = \mathbf{U}\hat{\mathbf{x}}$. The spectral graph convolution between a signal $\mathbf{x}$ and a filter $g$ is defined as:

$$\mathbf{x} *_{\mathcal{G}} g = \mathbf{U}g(\mathbf{\Lambda})\mathbf{U}^\top \mathbf{x}, \tag{1}$$

where $g(\mathbf{\Lambda})$ is a learnable filter function applied in the spectral domain.

The GCN [19] can be interpreted as a spectral GNN that employs a first-order Chebyshev polynomial to approximate the filter function, yielding $g(\mathbf{\Lambda}) = (\mathbf{I} + \mathbf{\Lambda})^{-1}$. This corresponds to a low-pass filter, which suppresses high-frequency components and retains smooth signal patterns across the graph [15].

**Graph Heterophily.** Many GNN models implicitly assume homophily, where connected nodes are expected to have similar features or labels [15]. However, in practice, numerous real-world graphs violate this assumption and instead exhibit high heterophily—connections between nodes of differing labels or characteristics [27]. To quantify this, we adopt the notion of node-level heterophily [11], defined as follows:

**Definition 1** (Node Heterophily). *Let $\mathcal{N}(v_i) = \{v_j \in \mathcal{V} : (v_i, v_j) \in \mathcal{E}\}$ denote the neighbors of node $v_i$, and let $y_i$ denote its label. The node heterophily $h_i$ for node $v_i$ is defined as:*

$$h_i = \frac{|\{v_j \in \mathcal{N}(v_i) : y_j \neq y_i\}|}{|\mathcal{N}(v_i)|}. \tag{2}$$

The heterophily degree vector $\mathbf{h} = (h_0, \ldots, h_{n-1}) \in \mathbb{R}^n$ captures the heterophily level of each node, and the overall heterophily of the graph is computed as the average: $\frac{1}{n}\sum_{i=0}^{n-1} h_i$.

High-heterophily graphs are common in domains such as social networks, citation graphs, and web hyperlinks. Benchmark datasets including *Texas*, *Squirrel*, *Chameleon*, *Cornell*, *Wisconsin*, and *Actor* [28, 29, 30] are representative examples. In such settings, conventional GCN-based message passing, which aggregates information from neighboring nodes, often leads to feature smoothing that can obscure useful class-distinctive signals, resulting in suboptimal performance.

## 3 Theoretical Bound in Terms of Graph Heterophily, Graph Filter, and Prediction Performance

In this section, we address **Q1** by theoretically analyzing the relationship among the graph filter, the heterophily degree of a graph $\mathcal{G}$, and the generalization performance of GNNs. We begin by demonstrating that the average spectral response of a filter and the heterophily level of the graph do not follow a simple monotonic relationship. This result, formalized in Proposition 1, motivates the need for adaptive filtering. We then establish that adaptive filters can, in principle, align closely with arbitrary label distributions (Proposition 2). Finally, we derive a generalization error bound (Theorem 1) that quantifies how graph heterophily and spectral filter design jointly affect learning performance. Full proofs of all results are provided in the Appendix C.

**Spectral Characterization of Heterophily.** While graph filters are naturally defined in the spectral domain, heterophily is typically measured in the spatial domain. To bridge this gap, we introduce a heterophily degree vector in the spectral domain via the graph Fourier transform.

**Definition 2** (Heterophily Degree Vector in Spectral Domain). *Let $\tilde{\mathbf{A}} = \mathbf{U}\mathbf{\Lambda}\mathbf{U}^\top$ be the eigendecomposition of the normalized adjacency matrix $\tilde{\mathbf{A}}$, where $\mathbf{U}$ contains the eigenvectors and $\mathbf{\Lambda}$ is the diagonal matrix of eigenvalues. Then the heterophily degree vector in the spectral domain $\hat{\mathbf{h}} \in \mathbb{R}^n$ is defined as:*

$$\hat{\mathbf{h}} = \mathbf{U}^\top \mathbf{h},$$

*where $\mathbf{h}$ is the heterophily degree vector (spatial) and $\hat{\mathbf{h}} = (\hat{h}_0, \ldots, \hat{h}_{n-1})$, $\hat{h}_i$ is heterophily of the $i$-th element in the spectral domain.*

**Non-Monotonicity of Frequency Response and Heterophily.** We next examine how the average spectral response of a graph filter relates to the heterophily in the spectral domain.

**Proposition 1.** *Let $\tilde{\mathbf{A}}$ be the normalized adjacency matrix with eigendecomposition $\tilde{\mathbf{A}} = \mathbf{U}\mathbf{\Lambda}\mathbf{U}^\top$, where $\lambda_0 \leq \cdots \leq \lambda_{n-1} = 1$. Then the average filter response is lower bounded by:*

$$\sum_{i=0}^{n-1} \frac{g(\lambda_i)}{n} \geq \frac{\sum_{i=0}^{n-1} \log |\hat{h}_i|}{n \left( \log \sum_{i=0}^{n-1} g(\lambda_i) |\hat{h}_i| - \log \sum_{i=0}^{n-1} g(\lambda_i) \right)}. \tag{3}$$

This result reveals that the average filter response $\frac{1}{n} \sum_i g(\lambda_i)$ and the average heterophily in the spectral domain $\frac{1}{n} \sum_i \hat{h}_i$ do not follow a monotonic relationship. As shown in Figure 1, graphs with low overall heterophily in the spatial domain exhibit strong responses in mid-frequency bands, contradicting the assumption that low-pass filters are always optimal in such cases.

**Need for Adaptive Filtering.** The non-monotonic and graph-dependent nature of heterophily–frequency interactions suggests that fixed filters (e.g., purely low-pass or high-pass) are suboptimal. To accommodate diverse frequency needs, we consider an adaptive polynomial filter:

$$g(\mathbf{\Lambda}) = \sum_{k=1}^{K} \sigma(\mathbf{w}_k \odot \mathbf{\Lambda}^k), \tag{4}$$

where $\sigma$ is an activation function, $\mathbf{w}_k \in \mathbb{R}^n$ are learnable weights, and $\odot$ denotes the Hadamard product. GCN [19] is a special case where $g(\mathbf{\Lambda}) = (\mathbf{I} + \mathbf{\Lambda})^{-1}$ corresponds to a low-pass filter [15].

We next show that such adaptive filters can match any label pattern in the frequency domain.

**Proposition 2.** *Let $\mathbf{Y} = (y_1, \ldots, y_n)$ be the label vector with $y_i \in \{0, \ldots, C-1\}$, and let $\mathbf{\Lambda}$ be the eigenvalue matrix of $\tilde{\mathbf{A}}$, assuming all eigenvalues are nonzero. Then there exist weights $\{\mathbf{w}_k\}_{k=1}^{K}$ such that:*

$$Align(g(\mathbf{\Lambda}), \mathbf{Y}) = 1,$$

*where $g(\mathbf{\Lambda}) = \sum_{k=1}^{K} \sigma(\mathbf{w}_k \odot \mathbf{\Lambda}^k)$, $\sigma(0) = 0$, and $Align(\cdot, \cdot)$ denotes cosine similarity.*

This proposition highlights the expressive power of adaptive filters to align with the target signal, particularly under varying heterophily.

**Generalization Error Bound.** We now quantify how filter design and heterophily in the spectral domain affect generalization error.

**Theorem 1.** *Consider a binary classification task on a graph $\mathcal{G}$ with $n$ nodes. Let $\mathbf{X} = (\mathbf{X}_0, \mathbf{X}_1)$ be the filtered node features for nodes belonging to class 0 and class 1, respectively. $\mathbf{Y} = (\mathbf{y}_0, \mathbf{y}_1)$ be the label indicators for classes 0 and 1. Let $\delta, \eta$ be the spectral coefficients of the label and feature differences, respectively. Then the error is upper bounded as:*

$$\overline{Er}(\mathbf{X}, \mathbf{Y}) \leq c_1 - \frac{\min_{i \in \mathcal{I}_{g,\delta,\eta}} \psi_{\frac{1}{g(1-\lambda_i)\delta_i}}(\eta_i) \cdot \delta_i \sum_{i \in \mathcal{I}_{\delta,\bar{\eta}}} \log |\hat{h}_i|}{2n \log \sum_{i \in \mathcal{I}_{\delta,\bar{\eta}}} g(1-\lambda_i)|\hat{h}_i| - 2n \log \sum_{i \in \mathcal{I}_{\delta,\bar{\eta}}} g(1-\lambda_i)}, \tag{5}$$

*where $c_1$ is a constant, $\mathcal{I}_\delta = \{i \mid \delta_i \neq 0\}$, and $\psi(x) = \min\{\max\{x, -1\}, 1\}$.*

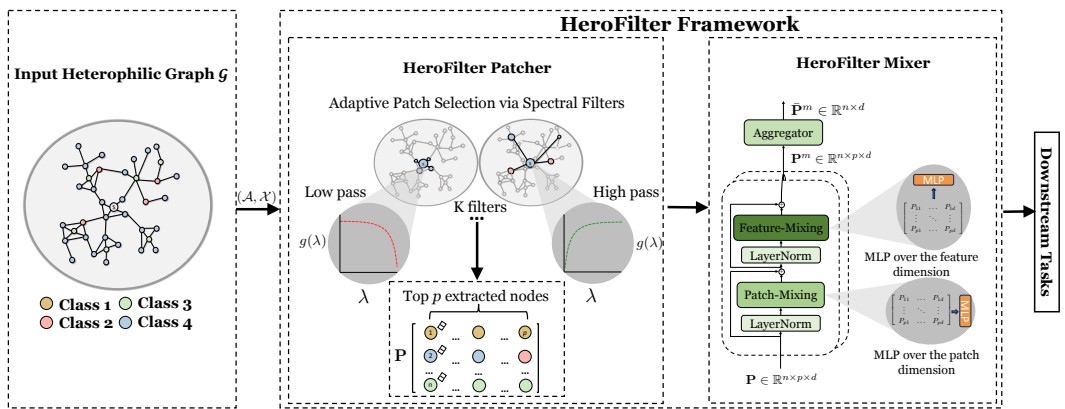

Figure 3: Overview of the HEROFILTER framework, consisting of (1) the HEROFILTER Patcher, which adaptively selects spectrally relevant neighbors using learned filters, and (2) the HEROFILTER Mixer, which aggregates patch features across nodes and dimensions.

Theorem 1 formally characterizes how filter choice and heterophily in the spectral domain influence prediction performance. To explore this relationship, we synthesized nine graphs with heterophily degrees ranging from 0.1 to 0.9 as shown in Figure 2. For each graph, we employed five learnable spectral filters passing specific eigenvalue segments: 0–0.4, 0.4–0.8, 0.8–1.2, 1.2–1.6, and 1.6–2.0. Our synthetic experiments demonstrate that different frequency regions contribute variably to accuracy depending on the heterophily level. This further supports the need for flexible, graph-specific filtering strategies.

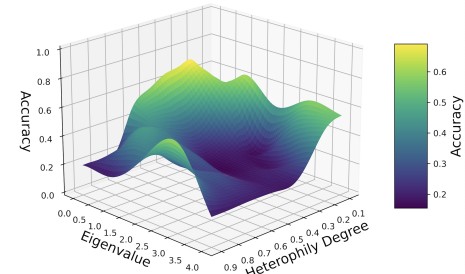

Figure 2: An interpolated figure demonstrating the relationship between filtered eigenvalues, heterophily degree, and accuracy.

# 4 HEROFILTER Framework

The theoretical analysis in Section 3 highlights two key limitations of existing GNN approaches when applied to heterophilic graphs. First, the spectral behavior of real-world graphs with varying heterophily is complex and non-monotonic, rendering uniform low-pass or high-pass filters ineffective. Second, different frequency bands contribute differently to prediction performance depending on the graph's heterophily structure. These insights motivate the design of HEROFILTER, a graph learning framework that adaptively aligns its spectral processing with the graph's heterophily profile.

HEROFILTER consists of two core modules: (1) The **HEROFILTER Patcher**, which dynamically selects spectrally relevant neighbors for each node by learning adaptive polynomial filters. This step enables the model to attend to contextually important nodes across spectral bands, beyond simple local neighborhoods, and (2) The **HEROFILTER Mixer**, which processes each node's patch using a dual-axis MLP architecture that mixes across both spatial (patch) and feature dimensions. This ensures that the model can effectively combine diverse signals arising from the selected patches.

Together, these components allow HEROFILTER to flexibly adapt its receptive field and frequency sensitivity, providing an interpretable and scalable mechanism for handling graphs with varying or mixed heterophily.

## 4.1 HEROFILTER Patcher

The goal of the patcher is to construct, for each node, a set of contextually informative neighbors that reflect spectral, rather than purely topological, similarity. Traditional GNNs aggregate over immediate neighbors, implicitly assuming local homophily. This assumption breaks down in heterophilic settings.

Thus, we propose to learn a filter $g(\mathbf{\Lambda})$ that scores nodes based on their spectral alignment, enabling a more adaptive notion of neighborhood.

We define the filter using a learnable polynomial function over the eigenvalues of the normalized adjacency matrix:

$$g(\mathbf{\Lambda}) = \sum_{k=1}^{K} \sigma(\mathbf{w}_k \odot \mathbf{\Lambda}^k), \tag{6}$$

where $\sigma$ is a non-linear activation function and $\mathbf{w}_k$ are learnable frequency-specific weights. This parameterization allows the model to selectively emphasize different frequency bands based on training data, consistent with our theoretical observation that fixed low/high-pass filters are suboptimal.

To apply this filter in the graph domain, we transform it via the graph Fourier basis $\mathbf{U}$, resulting in the node-node relevance matrix:

$$\mathbf{R} = \mathbf{U} \left( \sum_{k=1}^{K} \mathbf{w}_k \odot \mathbf{\Lambda}^k \right) \mathbf{U}^\top. \tag{7}$$

Here, $\mathbf{R}_{ij}$ measures the spectral relevance of node $j$ to node $i$ under the learned filter. For each node, we sort all other nodes by their relevance scores and select the top-$p$ most aligned ones to form the patch. Mathematically:

$$\phi(\tilde{\mathbf{A}}) = \text{top-}p_{\text{col}}\left(\mathbf{R}\right). \tag{8}$$

The resulting patches are structurally diverse and frequency-aware, allowing the model to capture high-order or cross-cluster interactions often present in heterophilic graphs.

For each selected patch, we extract the corresponding feature vectors from $\mathbf{X}$, yielding a patch tensor:

$$\mathbf{P}_v = \mathbf{X}[\text{top-}p(\mathbf{R}_v)], \quad \mathbf{P} \in \mathbb{R}^{n \times p \times d}, \tag{9}$$

where $p$ is the patch size and $d$ is the feature dimension. These patches now serve as the input for the next step of the model.

## 4.2 HEROFILTER Mixer

Once patches are constructed, we need to aggregate and transform them into useful node-level representations. Unlike traditional GNNs that rely on fixed aggregation schemes (e.g., mean, sum), our objective is to allow richer, learnable interactions both across patch elements (spatial mixing) and across features (feature mixing). This is motivated by the observation that information relevant to classification may lie in combinations of both positional and attribute-level patterns.

We adapt the MLP-Mixer architecture to this setting by defining two mixing layers:

**Patch-Mixing Layer.** For each node's patch $\mathbf{P}_v \in \mathbb{R}^{p \times d}$, we apply an MLP across the patch dimension:

$$\hat{\mathbf{P}}_v = \text{MLP}(\text{LayerNorm}(\mathbf{P}_v^\top))^\top. \tag{10}$$

This operation allows the model to reason over relationships between the selected nodes in the patch, independently for each feature. It enables the capture of role-based, cross-hop, or structurally asymmetric dependencies.

**Feature-Mixing Layer.** Next, we apply an MLP across the feature dimension:

$$\tilde{\mathbf{P}}_v = \text{MLP}(\text{LayerNorm}(\hat{\mathbf{P}}_v)). \tag{11}$$

This allows the model to learn joint representations over node features, enabling flexible composition and transformation of feature information within each patch.

The output of the mixer is a tensor $\tilde{\mathbf{P}} \in \mathbb{R}^{n \times p \times d}$. We then apply a global aggregation function (e.g., mean, sum, or flatten) over the patch dimension to produce a final node representation. This aggregated output is passed to a classification layer.

### 4.3 Why Mixing Both Dimensions

Patch-mixing captures the heterophily in node context by modeling which other nodes influence the representation of a given node. Feature-mixing captures intra-node complexity by transforming raw input features into abstract representations. Therefore, the combination is essential: patch mixing enables adaptivity to structural irregularities (e.g., in heterophilic regions), while feature mixing enables expressive node-level reasoning.

### 4.4 Fast-HEROFILTER Patcher

Although the HEROFILTER Patcher is expressive, its reliance on eigen decomposition can be prohibitive for large-scale graphs. To improve scalability, we propose Fast-HEROFILTER, an efficient variant that approximates the patcher using iterative proximity ranking, inspired by personalized PageRank and diffusion-based similarity.

We define the following objective for each node $v$:

$$\mathcal{J}(\mathbf{r}_v) = c\mathbf{r}_v^\top(\mathbf{I} - \tilde{\mathbf{A}})\mathbf{r}_v + (1 - c)\|\mathbf{r}_v - \mathbf{e}_v\|_2^2, \tag{12}$$

where $\mathbf{r}_v$ is a ranking vector and $\mathbf{e}_v$ is a one-hot vector. This objective balances global smoothness and local personalization.

Minimizing $\mathcal{J}(\mathbf{r}_v)$ yields the recurrence:

$$\mathbf{r}_v = c\,\tilde{\mathbf{A}}\,\mathbf{r}_v + (1 - c)\,\mathbf{e}_v, \tag{13}$$

which has the closed-form solution:

$$\mathbf{r}_v = (1 - c)(\mathbf{I} - c\,\tilde{\mathbf{A}})^{-1}\mathbf{e}_v.$$

To avoid matrix inversion, we approximate this with a truncated Neumann series:

$$\mathbf{r}_v \approx (1 - c)\sum_{k=0}^{K} c^k\,\tilde{\mathbf{A}}^k\,\mathbf{e}_v. \tag{14}$$

This expansion can be interpreted as diffusing information $K$ steps away from $v$, with attenuation $c^k$. Since $\tilde{\mathbf{A}}^k = \mathbf{U}\boldsymbol{\Lambda}^k\mathbf{U}^\top$, this operation also approximates spectral filtering.

Finally, we construct patches by selecting the top-$p$ entries in each $\mathbf{r}_v$, with the Fast-HEROFILTER patching rule:

$$\phi_{\text{fast}}(\tilde{\mathbf{A}}) = \text{top-}p_{\text{col}}([\mathbf{r}_1, \ldots, \mathbf{r}_n]). \tag{15}$$

Since $\mathbf{R}$ can be efficiently precomputed and patch extraction is fully parallelizable, this formulation enables scalable and adaptive patch construction, serving as a practical alternative to spectral filtering. Full pseudocode is provided in the Appendix D.

## 5 Experiment

We empirically evaluate HEROFILTER to validate the key claims made in this work: (1) that adaptive spectral filtering enables robust performance across both homophilic and heterophilic graphs; (2) that our proposed patching and mixing modules contribute significantly to model generalization; and (3) that HEROFILTER offers a scalable, interpretable, and architecture-agnostic foundation for future graph learning models. Due to space constraints, we include additional experiments (scalability, sensitivity, and runtime) in the Appendix G and Appendix H.

We evaluate HEROFILTER on 16 node classification benchmarks encompassing diverse graph heterophily degrees. The homophilic graphs include Cora, CiteSeer, PubMed, and OGBN-Arxiv, where neighboring nodes tend to share similar labels. The heterophilic graphs consist of Texas, Squirrel, Chameleon, Cornell, Wisconsin, Actor, Arxiv-Year, and Snap-Patents, which feature connections between dissimilar nodes and present greater challenges for conventional GNNs. Notably, Arxiv-Year and Snap-Patents also serve as large-scale datasets, with up to 3 million nodes and 14 million edges, enabling assessment of the model's scalability and efficiency in high-volume scenarios.

Table 1: Comparison of different methods in node classification task on homophilic graph datasets. We denote *red*, *green*, *blue* as the best, second best, and third best performance, respectively. OOM denotes "Out of Memory".

| Model | Cora | CiteSeer | PubMed | OGBN-Arxiv |
|---|---|---|---|---|
| $\frac{\sum h_i}{n}$ | 0.19 | 0.26 | 0.20 | 0.34 |
| MLP | $72.0 \pm 1.7$ | $71.8 \pm 1.7$ | $85.3 \pm 0.4$ | $49.7 \pm 0.6$ |
| GPRGNN | $86.2 \pm 1.1$ | $74.7 \pm 1.8$ | $87.6 \pm 0.5$ | $64.6 \pm 1.2$ |
| ChebNet | $85.5 \pm 1.1$ | $75.4 \pm 1.4$ | $87.7 \pm 0.5$ | OOM |
| ChebNetII | $84.0 \pm 1.2$ | $71.1 \pm 1.8$ | $86.2 \pm 0.3$ | $67.0 \pm 0.8$ |
| APPNP | $86.7 \pm 1.0$ | $74.5 \pm 1.2$ | $86.9 \pm 0.3$ | $62.4 \pm 1.4$ |
| GCNJKNet | $86.5 \pm 1.1$ | $75.2 \pm 1.5$ | $87.4 \pm 0.4$ | $65.1 \pm 0.6$ |
| GCN | $86.1 \pm 1.0$ | $74.3 \pm 1.2$ | $86.9 \pm 0.4$ | $64.9 \pm 0.5$ |
| GAT | $85.9 \pm 1.0$ | $74.7 \pm 1.3$ | $85.8 \pm 0.5$ | $65.7 \pm 0.6$ |
| GraphSage | $84.9 \pm 1.4$ | $73.4 \pm 1.0$ | $86.4 \pm 0.5$ | $64.7 \pm 0.7$ |
| FAGCN | $85.5 \pm 0.7$ | $74.8 \pm 1.5$ | $87.0 \pm 0.3$ | $64.1 \pm 0.2$ |
| H$_2$GCN | $86.5 \pm 1.2$ | $74.6 \pm 1.7$ | $87.7 \pm 0.4$ | OOM |
| BM-GCN | $86.1 \pm 1.1$ | $74.3 \pm 1.3$ | $86.8 \pm 1.1$ | OOM |
| BernNet | $84.9 \pm 1.7$ | $72.3 \pm 1.4$ | $87.6 \pm 0.4$ | OOM |
| G$^2$-GCN | $80.2 \pm 1.7$ | $70.2 \pm 1.4$ | $87.0 \pm 0.4$ | $62.6 \pm 1.3$ |
| GOAT | $84.0 \pm 0.8$ | $72.1 \pm 1.6$ | $87.1 \pm 0.8$ | $70.0 \pm 0.4$ |
| NAGphormer | $85.7 \pm 1.2$ | $73.5 \pm 1.2$ | $87.5 \pm 0.3$ | $68.1 \pm 0.2$ |
| PolyFormer | $83.2 \pm 2.5$ | $73.5 \pm 2.3$ | $87.5 \pm 0.2$ | $70.4 \pm 0.2$ |
| Exphormer | $84.7 \pm 0.5$ | $74.4 \pm 1.2$ | $87.6 \pm 0.3$ | $69.2 \pm 0.5$ |
| VCR-Graphormer | $83.8 \pm 2.1$ | $73.7 \pm 0.5$ | $87.7 \pm 0.4$ | $68.2 \pm 0.2$ |
| HEROFILTER(Ours) | $86.8 \pm 1.5$ | $75.0 \pm 1.2$ | $87.6 \pm 0.3$ | $70.5 \pm 0.4$ |

Table 2: Comparison of different methods in node classification task on heterophilic datasets. We denote *red*, *green*, *blue* as the best, second best, and third best performance, respectively. OOM denotes "Out of Memory".

| Model | Snap-Patents | Arxiv-Year | Texas | Squirrel | Chameleon | Cornell | Wisconsin | Actor |
|---|---|---|---|---|---|---|---|---|
| $\frac{\sum h_i}{n}$ | 0.93 | 0.78 | 0.89 | 0.78 | 0.77 | 0.89 | 0.84 | 0.76 |
| MLP | $31.0 \pm 0.1$ | $35.7 \pm 0.1$ | $74.4 \pm 5.6$ | $33.6 \pm 1.4$ | $49.4 \pm 1.6$ | $69.9 \pm 2.9$ | $80.4 \pm 3.7$ | $35.0 \pm 0.7$ |
| GPRGNN | $41.1 \pm 1.1$ | $45.2 \pm 0.1$ | $64.2 \pm 5.3$ | $30.4 \pm 2.0$ | $39.2 \pm 2.0$ | $59.1 \pm 4.4$ | $72.8 \pm 4.5$ | $30.9 \pm 0.7$ |
| ChebNet | OOM | $46.3 \pm 0.1$ | $74.4 \pm 5.8$ | $33.4 \pm 1.0$ | $49.4 \pm 1.7$ | $69.7 \pm 2.8$ | $77.9 \pm 3.0$ | $34.8 \pm 1.0$ |
| ChebNetII | OOM | $40.4 \pm 0.1$ | $82.1 \pm 5.6$ | $37.5 \pm 1.2$ | $52.2 \pm 1.5$ | $72.2 \pm 2.3$ | $83.8 \pm 2.2$ | $35.1 \pm 1.1$ |
| APPNP | $40.9 \pm 0.1$ | $44.1 \pm 0.6$ | $56.3 \pm 3.9$ | $27.8 \pm 0.7$ | $44.3 \pm 1.5$ | $42.9 \pm 3.7$ | $52.3 \pm 4.9$ | $28.4 \pm 0.6$ |
| GCNJKNet | OOM | $46.7 \pm 0.2$ | $57.8 \pm 2.0$ | $26.3 \pm 0.6$ | $42.2 \pm 2.1$ | $40.9 \pm 4.6$ | $49.6 \pm 4.8$ | $27.9 \pm 0.8$ |
| GCN | $46.1 \pm 0.1$ | $46.2 \pm 0.2$ | $58.7 \pm 2.8$ | $27.1 \pm 0.5$ | $41.4 \pm 1.7$ | $40.3 \pm 3.3$ | $49.4 \pm 3.9$ | $28.4 \pm 0.7$ |
| GAT | OOM | $47.1 \pm 0.1$ | $58.3 \pm 4.8$ | $28.7 \pm 0.9$ | $43.9 \pm 1.6$ | $46.3 \pm 4.3$ | $51.1 \pm 6.2$ | $28.9 \pm 0.5$ |
| GraphSage | $48.9 \pm 0.1$ | $48.1 \pm 0.2$ | $74.3 \pm 3.7$ | $36.2 \pm 0.8$ | $46.2 \pm 1.8$ | $69.1 \pm 3.5$ | $76.9 \pm 4.5$ | $34.6 \pm 0.7$ |
| FAGCN | OOM | $41.7 \pm 3.1$ | $65.4 \pm 2.8$ | $33.9 \pm 0.8$ | $43.2 \pm 0.9$ | $55.4 \pm 5.1$ | $65.7 \pm 4.3$ | $34.4 \pm 0.5$ |
| H$_2$GCN | OOM | $48.4 \pm 0.6$ | $81.1 \pm 5.2$ | $32.2 \pm 1.2$ | $54.0 \pm 1.1$ | $68.5 \pm 2.9$ | $78.0 \pm 3.1$ | $34.9 \pm 0.4$ |
| BM-GCN | OOM | OOM | $75.4 \pm 4.8$ | $34.2 \pm 0.9$ | $53.6 \pm 1.5$ | $62.3 \pm 4.3$ | $75.4 \pm 5.2$ | $34.7 \pm 0.5$ |
| BernNet | OOM | $38.2 \pm 0.2$ | $80.7 \pm 3.3$ | $38.7 \pm 0.8$ | $50.9 \pm 1.2$ | $71.8 \pm 2.8$ | $84.3 \pm 4.3$ | $35.4 \pm 0.6$ |
| G$^2$-GCN | OOM | $51.2 \pm 0.5$ | $81.3 \pm 3.7$ | $38.5 \pm 0.7$ | $52.9 \pm 1.3$ | $71.1 \pm 4.1$ | $84.1 \pm 4.4$ | $34.3 \pm 0.4$ |
| GOAT | $55.0 \pm 0.2$ | $53.5 \pm 0.2$ | $62.2 \pm 5.3$ | $35.4 \pm 1.2$ | $49.9 \pm 2.4$ | $52.4 \pm 4.1$ | $74.6 \pm 0.4$ | $34.2 \pm 0.9$ |
| NAGphormer | $54.8 \pm 0.1$ | $47.8 \pm 0.2$ | $69.1 \pm 6.7$ | $36.3 \pm 0.9$ | $47.6 \pm 1.2$ | $63.8 \pm 4.1$ | $76.7 \pm 4.0$ | $34.8 \pm 0.6$ |
| PolyFormer | OOM | $45.1 \pm 0.2$ | $63.5 \pm 5.4$ | $41.5 \pm 1.0$ | $54.9 \pm 1.7$ | $60.8 \pm 3.5$ | $75.8 \pm 3.9$ | $34.9 \pm 0.3$ |
| Exphormer | OOM | $47.0 \pm 0.2$ | $76.3 \pm 1.1$ | $32.9 \pm 0.5$ | $46.5 \pm 0.9$ | $63.5 \pm 1.5$ | $77.2 \pm 1.0$ | $34.7 \pm 0.6$ |
| VCR-Graphormer | OOM | $53.7 \pm 1.0$ | $64.2 \pm 4.0$ | $34.2 \pm 2.0$ | $56.9 \pm 1.7$ | $58.9 \pm 3.5$ | $53.1 \pm 9.2$ | $34.8 \pm 0.5$ |
| HEROFILTER(Ours) | $64.2 \pm 0.1$ | $54.6 \pm 0.1$ | $85.0 \pm 3.5$ | $47.7 \pm 2.2$ | $57.3 \pm 1.2$ | $72.5 \pm 3.8$ | $85.1 \pm 3.1$ | $37.8 \pm 0.7$ |

We compare against classical GNNs (GCN, GAT, GraphSAGE, ChebNet), spectral models (BernNet, GPRGNN), heterophily-aware methods (H$_2$GCN, BM-GCN, G$^2$GCN), and graph transformers (GOAT, NAGphormer, Exphormer, PolyFormer, VCR-Graphormer). Detailed dataset and model descriptions are provided in the Appendix E.

Table 3: Patchers on Cora and Chameleon.

| Patchers | Cora | Chameleon |
|---|---|---|
| Bandpass Filter | $71.3 \pm 0.8$ | $52.1 \pm 1.5$ |
| Heat Filter | $70.4 \pm 1.7$ | $47.7 \pm 1.7$ |
| Shared parameters | $73.0 \pm 1.3$ | $52.2 \pm 1.1$ |
| HEROFILTER | $\mathbf{77.3 \pm 0.6}$ | $\mathbf{57.3 \pm 1.2}$ |

Table 4: Random vs. ranked patch order.

| Dataset | Random | Ranked | Change |
|---|---|---|---|
| Cora | $71.4 \pm 1.1$ | $\mathbf{77.3 \pm 0.6}$ | ↑ **5.9%** |
| Citeseer | $60.7 \pm 1.4$ | $\mathbf{64.5 \pm 1.8}$ | ↑ **3.8%** |
| Squirrel | $35.4 \pm 1.2$ | $\mathbf{37.8 \pm 1.6}$ | ↑ **2.4%** |
| Chameleon | $51.0 \pm 1.2$ | $\mathbf{57.3 \pm 1.1}$ | ↑ **6.3%** |

Table 5: Accuracy on patch-induced graphs.

| Model | Cora | Chameleon |
|---|---|---|
| GCN | $76.1 \pm 2.3$ | $39.9 \pm 2.0$ |
| GCN-Patch | $74.1 \pm 1.0$ | $56.0 \pm 2.0$ |
| FAGCN | $73.2 \pm 0.7$ | $43.2 \pm 0.9$ |
| FAGCN-Patch | $74.8 \pm 0.4$ | $48.3 \pm 2.2$ |
| HEROFILTER | $\mathbf{77.3 \pm 0.6}$ | $\mathbf{57.3 \pm 1.2}$ |

Table 6: Indivdual component effectiveness in HEROFILTER.

| Ablation | Arxiv-Year | Snap-Patents |
|---|---|---|
| HEROFILTER | $\mathbf{54.66 \pm 0.28}$ | $\mathbf{65.05 \pm 0.04}$ |
| w/ Patch-Mixing | $54.49 \pm 0.04$ | $64.97 \pm 0.05$ |
| w/ Feature-Mixing | $53.85 \pm 0.20$ | $63.00 \pm 0.01$ |
| Both Removed | $52.81 \pm 0.10$ | $62.95 \pm 0.09$ |

## 5.1 Overall Performance

Tables 1 and 2 summarize classification accuracy across all datasets. Three core insights emerge:

*(1) HEROFILTER achieves strong performance across both homophilic and heterophilic regimes.* On homophilic graphs, HEROFILTER matches or exceeds state-of-the-art performance (e.g., 70.5% on OGBN-Arxiv). More importantly, HEROFILTER substantially outperforms baselines on heterophilic datasets. For instance, on Squirrel and Snap-Patents, HEROFILTER improves over the next-best model by 9.0% and 9.2%, respectively. This validates the theoretical claim that a fixed low-pass or high-pass filter is insufficient for generalization across heterophily levels and underscores the effectiveness of adaptive spectral filtering.

*(2) HEROFILTER remains effective on large and noisy graphs.* While several baselines fail on large graphs (marked OOM), HEROFILTER maintains strong accuracy (e.g., 64.2% on Snap-Patents) without sacrificing runtime scalability. This highlights the value of patch-based processing and Fast-HEROFILTER for memory efficiency—an increasingly critical concern in modern GNN deployments.

*(3) HEROFILTER bridges gaps between GNNs, spectral methods, and transformers.* Transformer models with strong inductive biases underperform on small or irregular graphs (e.g., Texas, Wisconsin). In contrast, HEROFILTER's simpler design generalizes well without architectural complexity. Compared to BernNet or GPRGNN, HEROFILTER does not require fine-tuned frequency designs and yet achieves superior results, highlighting its robustness and ease of use.

## 5.2 Ablation Studies

To better understand HEROFILTER's internal design, we conduct controlled experiments analyzing the contributions of the Patcher, Mixer, and positional structure.

**Effectiveness of Patchers.** We compare HEROFILTER's adaptive polynomial patcher against several alternatives: heat filter, bandpass filter, and a shared-parameter variant. As shown in Table 3, our method consistently outperforms others, especially on heterophilic graphs (e.g., 57.3% on Chameleon). This highlights the importance of a learnable and flexible filter form, consistent with our theoretical insights from Section 3. While bandpass and heat filters capture fixed spectral bands, they lack the adaptability needed for general graphs.

**Importance of Patch Order.** We assess the role of node order within each patch. As shown in Table 4, randomly shuffling the patch degrades performance on all datasets, with up to 6.3% accuracy loss on Chameleon. This demonstrates that patch structure encodes meaningful spectral and positional information, and that the model effectively leverages this structure, an important design signal for future permutation-sensitive architectures.

**Patch-Induced Graphs.** To isolate the value of the patcher, we create graphs by linking each node only to its selected patch neighbors, then apply existing GNNs (GCN, FAGCN). Results in Table 5

show significant performance gains, especially for GCN on Chameleon (+16%), validating that HEROFILTER Patcher successfully surfaces informative neighbors even for GNNs not explicitly tuned to heterophily. Conversely, slight degradation on Cora for GCN confirms that fixed low-pass assumptions may conflict with spectrally diverse patches.

**Component-Level Analysis.** We remove or replace individual layers in the HEROFILTER Mixer to evaluate their contribution (Table 6). Removing the patch-mixing or feature-mixing layers causes consistent drops in performance across datasets, confirming their complementary roles. Notably, the patch-mixing layer provides the largest standalone gain (e.g., +2.02% on Snap-Patents), suggesting its importance in modeling structural diversity within patches.

## 6 Related Work

**Learning on Heterophilic Graphs.** Traditional GNNs, including GCN [19], GAT [20], and APPNP [31], are grounded in the homophily assumption, neighboring nodes tend to share similar labels. However, real-world networks often violate this assumption, exhibiting *heterophily* where connected nodes belong to different classes [32, 27]. This has motivated a range of models that enhance GNN performance under heterophily. $H_2$GCN [32] and BM-GCN [33] extend message passing to multi-hop neighborhoods or learn block-level compatibility. Other methods like FAGCN [34] and BernNet [22] leverage spectral insights, using learnable filters to balance low- and high-frequency information. ChebNetII [35] revisits Chebyshev polynomials for deeper spectral modeling, while $G^2$GCN [36] introduces gradient gating mechanisms. These works highlight the importance of high-frequency components in heterophilic settings but often rely on heuristic assumptions or task-specific filter tuning.

**Spectral GNNs and Frequency-Adaptive Filters.** A complementary line of work focuses on understanding GNNs from the spectral perspective. The view of GNNs as graph filters in the spectral domain, applying transformations $g(\mathbf{\Lambda})$ via eigen decomposition of the Laplacian, has been formalized in studies such as [15, 37]. Spectral GNNs such as ChebNet [21] and GPRGNN [37] design polynomial filters to capture information across the graph spectrum. While these methods enhance theoretical interpretability and offer flexibility, most assume a global, fixed filter shape (e.g., low-pass or band-pass), which limits adaptivity. As recent theoretical works note [17, 16], the relationship between graph heterophily and optimal spectral response is non-monotonic and dataset-dependent, suggesting the need for instance-level adaptivity in filter design.

**Graph Transformers and Global Attention.** Transformer architectures have been adapted to graphs to address the limitations of locality in message passing. Approaches such as NAGphormer [38], GOAT [39], Exphormer [40], VCR-Graphormer [41], and PolyFormer [42] integrate global attention mechanisms, often augmented with positional encodings or spectral bias terms. While effective on large-scale and heterophilic graphs, these models tend to be computationally expensive and architecturally complex. Moreover, their reliance on learned positional encodings can make generalization brittle, especially on smaller graphs or graphs with evolving structure.

Our proposed HEROFILTER bridges the gap between the spectral flexibility of filter-based GNNs and the architectural simplicity of transformer-free encoders. Instead of hard-code frequency biases or depending on hand-tuned spectral forms, HEROFILTER introduces an adaptive polynomial filter that dynamically aligns its frequency response with the graph's heterophily structure. Theoretical results provide the first formal link between graph heterophily, spectral filter response, and generalization error. This connection not only strengthens empirical observations from prior work [27, 22], but also introduces a framework for future frequency-aware and architecture-agnostic GNN design.

## 7 Conclusion

In this paper, we make three key contributions: (1) a theoretical analysis that, for the first time, formally connects graph heterophily, spectral filter response, and generalization error—challenging the prevailing assumption of monotonic filter-heterophily correlation; (2) a modular architecture that integrates adaptive polynomial filters with a lightweight MLP-Mixer backbone, enabling interpretable and efficient spectral reasoning across diverse graph structures; and (3) extensive empirical validation across 16 benchmark datasets, where HEROFILTER consistently outperforms state-of-the-art GNNs and graph transformers on both homophilic and heterophilic graphs, including large-scale settings.

## Acknowledgements

We thank the anonymous reviewers for their constructive comments. This work is supported by the MIT-IBM Watson AI Lab, National Science Foundation under Award No. IIS-2339989 and No. 2406439, DARPA under contract No. HR00112490370 and No. HR001124S0013, U.S. Department of Homeland Security under Grant Award No. 17STCIN00001-08-00, Amazon-Virginia Tech Initiative for Efficient and Robust Machine Learning, Amazon AGI Team, Amazon AWS, Google, Cisco, 4-VA, Commonwealth Cyber Initiative, National Surface Transportation Safety Center for Excellence, UIUC AICE Center, and Virginia Tech. The views and conclusions are those of the authors and should not be interpreted as representing the official policies of the funding agencies or the government. We thank Professor Julian Shun (MIT) for his insightful discussions and feedback that improved this manuscript.

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

# A Appendix Content

- Appendix B: Key symbols and notations in this paper.
- Appendix C: Detailed proofs of Proposition 1, Proposition 2, and Theorem 1.
- Appendix D: Algorithm descriptions including HERoFILTER Framework and Fast-HERoFILTER Framework.
- Appendix E: Dataset statistics, including node count, edge count, number of classes, feature dimensions, and heterophily scores for both homophilic and heterophilic graphs. Details of baseline models used for comparison and implementation details.
- Appendix F: Parameter settings and tuning procedures, including hyperparameter search spaces.
- Appendix G: Scalability and runtime analysis of Fast-HERoFILTER and baseline models showing the inference speed of each model.
- Appendix H: Parameter sensitivity studies showing the impact of patch size and filter numbers.

# B Symbols and Notations

Table 7: Symbols and notations.

| Symbol | Description |
|---|---|
| $\mathcal{G}$ | Graph |
| $\mathcal{V}$ | Set of vertices (nodes) of the graph $\mathcal{G}$ |
| $\mathcal{E}$ | Set of edges of the graph $\mathcal{G}$ |
| $\mathbf{X}$ | Node feature matrix of $\mathcal{G}$. |
| $n$ | Number of nodes in the graph $\mathcal{G}$ |
| $\mathbf{L}, \mathbf{A}$ | Laplacian matrix and adjacency matrix of $\mathcal{G}$ |
| $\tilde{\mathbf{L}}, \tilde{\mathbf{A}}$ | Normalized Laplacian matrix and adjacency matrix of $\mathcal{G}$ |
| $\mathcal{N}(v)$ | Set of neighbors of node $v$ |
| $g(\mathbf{\Lambda})$ | A spectral graph filter on eigenvalue matrix $\mathbf{\Lambda}$ |

# C Proofs of the Theorems

**Proposition 1.** *Let $\tilde{\mathbf{A}}$ be the normalized adjacency matrix with eigendecomposition $\tilde{\mathbf{A}} = \mathbf{U}\mathbf{\Lambda}\mathbf{U}^\top$, where $\lambda_0 \leq \cdots \leq \lambda_{n-1} = 1$. Then the average filter response is lower bounded by:*

$$\sum_{i=0}^{n-1} \frac{g(\lambda_i)}{n} \geq \frac{\sum_{i=0}^{n-1} \log |\hat{h}_i|}{n \left( \log \sum_{i=0}^{n-1} g(\lambda_i)|\hat{h}_i| - \log \sum_{i=0}^{n-1} g(\lambda_i) \right)}. \tag{3}$$

*Proof.* According to the weighted AM-GM inequality [43], we have

$$\frac{g(\lambda_0)}{\sum_{0 \leq i \leq n-1} g(\lambda_i)}|\hat{h}_0| + \cdots + \frac{g(\lambda_{n-1})}{\sum_{0 \leq i \leq n-1} g(\lambda_i)}|\hat{h}_{n-1}|$$
$$\geq |\hat{h}_0|^{\frac{g(\lambda_0)}{\sum_{0 \leq i \leq n-1} g(\lambda_i)}} \cdot \ldots \cdot |\hat{h}_{n-1}|^{\frac{g(\lambda_{n-1})}{\sum_{0 \leq i \leq n-1} g(\lambda_i)}}. \tag{16}$$

As we have valid polynomial filer $g(\lambda_i) \in [0,1]$ for all $i$ [44], then

$$\frac{\sum_{0 \leq i \leq n-1} g(\lambda_i)|\hat{h}_i|}{\sum_{0 \leq i \leq n-1} g(\lambda_i)} \geq |\hat{h}_0|^{\frac{1}{\sum_{0 \leq i \leq n-1} g(\lambda_i)}} \cdot \ldots \cdot |\hat{h}_{n-1}|^{\frac{1}{\sum_{0 \leq i \leq n-1} g(\lambda_i)}}$$
$$= \prod_{0 \leq i \leq n-1} |\hat{h}_i|^{\frac{1}{\sum_{0 \leq k \leq n-1} g(\lambda_k)}}. \tag{17}$$

Since both sides of the inequality are greater than 0, we take the logarithm

$$\log\left(\frac{\sum_{0\leq i\leq n-1} g(\lambda_i)|\hat{h}_i|}{\sum_{0\leq i\leq n-1} g(\lambda_i)}\right) \geq \frac{1}{\sum_{0\leq k\leq n-1} g(\lambda_k)} \sum_{0\leq i\leq n-1} \log|\hat{h}_i|. \tag{18}$$

Hence, it completes the proof. $\qquad\square$

**Proposition 2.** *Let $\mathbf{Y} = (y_1, \ldots, y_n)$ be the label vector with $y_i \in \{0, \ldots, C-1\}$, and let $\mathbf{\Lambda}$ be the eigenvalue matrix of $\tilde{\mathbf{A}}$, assuming all eigenvalues are nonzero. Then there exist weights $\{\mathbf{w}_k\}_{k=1}^K$ such that:*

$$Align(g(\mathbf{\Lambda}), \mathbf{Y}) = 1,$$

*where $g(\mathbf{\Lambda}) = \sum_{k=1}^K \sigma(\mathbf{w}_k \odot \mathbf{\Lambda}^k)$, $\sigma(0) = 0$, and Align$(\cdot, \cdot)$ denotes cosine similarity.*

*Proof.* To match $g(\mathbf{\Lambda})$ with $\mathbf{Y}$, we require $g(\mathbf{\Lambda}) = c\mathbf{Y}$, where $c \neq 0$ is a scalar. This requires:

$$\sum_{k=1}^K \sigma\left(w_{k,i} \cdot \lambda_i^k\right) = \alpha y_i \quad \forall i \in \{1, 2, \ldots, n\}. \tag{19}$$

If $\mathbf{Y}_i = 0$, then we need $g_i(\mathbf{\Lambda}) = 0, \forall i \in \{1, \ldots, n\}$. Let $w_{k,i} = 0$ for all $k$, since $\sigma(0) = 0$, the conclusion holds.

If $\mathbf{Y}_i > 0$. Case 1: $\sigma(x)$ is bounded. Suppose $\sigma(x) \in [0, M]$ for some $M > 0$. Since $\mathbf{Y}_i \in [0, C-1]$, we have $\frac{c\mathbf{Y}_i}{K} \in [0, M]$ by choosing $c$ large enough. Since $\sigma$ is continuous and monotonic, we can always find $w_{k,i}$ to satisfy $\sigma(w_{k,i}(\Lambda_i)^k) = \frac{c\mathbf{Y}_i}{K}$.

Case 2: $\sigma(x)$ is unbounded. In this case, no scaling constraints are required. The left term can be arbitrarily large if $w_{k,i}$ is chosen large enough. So we can find $w_{k,i}$, s.t. $\sigma(w_{k,i}(\Lambda_i)^k) = \frac{c\mathbf{Y}_i}{K}$.

Thus, we have

$$g_i(\mathbf{\Lambda}) = \sum_{k=1}^K \sigma(w_{k,i}(\Lambda_i)^k) = c\mathbf{Y}_i, \tag{20}$$

for all $i \in \{1, \ldots, n\}$. Then we have

$$g(\mathbf{\Lambda}) = c\mathbf{Y}. \tag{21}$$

The cosine similarity is

$$\cos(g(\mathbf{\Lambda}), \mathbf{Y}) = \frac{g(\mathbf{\Lambda}) \cdot \mathbf{Y}}{\|g(\mathbf{\Lambda})\|\|\mathbf{Y}\|} = 1. \tag{22}$$

Hence, it completes the proof. $\qquad\square$

**Theorem 1.** *Consider a binary classification task on a graph $\mathcal{G}$ with $n$ nodes. Let $\mathbf{X} = (\mathbf{X}_0, \mathbf{X}_1)$ be the filtered node features for nodes belonging to class 0 and class 1, respectively. $\mathbf{Y} = (\mathbf{y}_0, \mathbf{y}_1)$ be the label indicators for classes 0 and 1. Let $\delta, \eta$ be the spectral coefficients of the label and feature differences, respectively. Then the error is upper bounded as:*

$$\overline{Er}(\mathbf{X}, \mathbf{Y}) \leq c_1 - \frac{\min\limits_{i\in\mathcal{I}_{g,\delta,\eta}} \psi_{\frac{1}{g(1-\lambda_i)\delta_i}}(\eta_i) \cdot \delta_i \sum\limits_{i\in\mathcal{I}_{\delta,\tilde{\eta}}} \log|\hat{h}_i|}{2n\log \sum\limits_{i\in\mathcal{I}_{\delta,\tilde{\eta}}} g(1-\lambda_i)|\hat{h}_i| - 2n\log \sum\limits_{i\in\mathcal{I}_{\delta,\tilde{\eta}}} g(1-\lambda_i)}, \tag{5}$$

*where $c_1$ is a constant, $\mathcal{I}_\delta = \{i \mid \delta_i \neq 0\}$, and $\psi(x) = \min\{\max\{x, -1\}, 1\}$.*

*Proof.* Let $\psi$ be the clamp function defined as

$$\psi(x) \triangleq \min\{\max\{x, -1\}, 1\} = \begin{cases} 1 & x > 1 \\ x & -1 < x < 1 \\ -1 & x < -1 \end{cases}, \tag{23}$$

$$d(x, \psi(x)) \triangleq \left(\frac{1}{1+e^x} - y\right)^2 - \left(\frac{1}{1+e^{\psi(x)}} - y\right)^2$$

$$\in \begin{cases} [-(\frac{1}{1+e})^2, 0] & x > 1, y = 0 \\ [0, (\frac{1}{1+c})^2] & x > 1, y = 1 \\ [0, (\frac{1}{1+e})^2] & x < -1, y = 0 \\ [-(\frac{1}{1+e})^2, 0] & x < -1, y = 1 \end{cases} \tag{24}$$

$$\leq \frac{1}{(1+e)^2},$$

and for $x \in [-1, 1]$, the first-order Taylor expansion of $\frac{1}{1+e^x}$ is $\frac{1}{2} - \frac{1}{4}x$. Denote $R(x)$ as the remainder term, that is, $R(x) = \frac{1}{1+e^x} - \frac{1}{2} + \frac{1}{4}x$. since

$$\left(\frac{1}{(1+e^x)^2}\right)''' = -\frac{e^x\left(-4e^x + e^{2x} + 1\right)}{(1+e^x)^4} \leq \left(\frac{1}{(1+e^x)^2}\right)'''\bigg|_{x=0} = \frac{1}{8}, \tag{25}$$

we have

$$|R(x)| \leq \max\left|\left(\frac{1}{(1+e^x)^2}\right)'''\right| \frac{|x|^3}{3!} = \frac{|x|^3}{48}. \tag{26}$$

Therefore,

$$\left(\frac{1}{1+e^x} - y\right)^2 = \left(\frac{1}{1+e^{\psi(x)}} - y\right)^2 + d(x, \psi(x))$$

$$= \left(\frac{1}{2} - \frac{1}{4}\psi(x) - y + R(\psi(x))\right)^2 + \frac{1}{(1+e)^2}$$

$$= \left(\frac{1}{2} - \frac{1}{4}\psi(x) - y\right)^2 + (R(\psi(x)))^2 + 2R(\psi(x))\left(\frac{1}{2} - \frac{1}{4}\psi(x) - y\right) + \frac{1}{(1+e)^2} \tag{27}$$

$$= \left(\frac{1}{2} - \frac{1}{4}\psi(x) - y\right)^2 + \left(\frac{|\psi(x)|^3}{48}\right)^2 + \frac{|\psi(x)|^3}{24}\left|\frac{1}{2} - \frac{1}{4}\psi(x) - y\right| + \frac{1}{(1+e)^2}$$

$$\leq \left(\frac{1}{2} - y\right)^2 - \frac{(1-2y)\psi(x)}{4} + \frac{\psi(x)^2}{16} + \frac{|\psi(x)|^6}{2304} + \frac{|\psi(x)|^3}{24}\left(\frac{1}{4}|\psi(x)| + \frac{1}{2}\right) + \frac{1}{(1+e)^2}$$

$$\leq \frac{1}{4} - \frac{(1-2y)\psi(x)}{4} + \frac{\psi(x)^2}{16} + \frac{|\psi(x)|^3}{48} + \frac{\psi(x)^4}{96} + \frac{|\psi(x)|^6}{2304} + \frac{1}{(1+e)^2}.$$

According to the above conclusion,

$$Er\left(\mathbf{X}_0, \mathbf{y}_0\right) = \sum_l \left(\frac{1}{1+e^{g(I-\tilde{\mathbf{A}})(\mathbf{x}_{1l} - \mathbf{x}_{0l})}} - \mathbf{y}_{0l}\right)^2$$

$$\leq \frac{n}{4} - \frac{1}{4}(\mathbf{y}_1 - \mathbf{y}_0)^\top \psi(\mathbf{z}) + \frac{\|\psi(\mathbf{z})\|_2^2}{16} + \frac{\|\psi(\mathbf{z})\|_3^3}{48} + \frac{\|\psi(\mathbf{z})\|_4^4}{96} + \frac{\|\psi(\mathbf{z})\|_6^6}{2304} + \frac{n}{(1+e)^2}, \tag{28}$$

$\mathbf{z} = g(I - \tilde{\mathbf{A}})(\mathbf{X}_1 - \mathbf{X}_0)_l$, noting that $C \leq \|\psi(\mathbf{z})\|_6^6 \leq \|\psi(\mathbf{z})\|_4^4 \leq \|\psi(\mathbf{z})\|_3^3 \leq \|\psi(\mathbf{z})\|_2^2 \leq n$, then we have

$$Er\left(\mathbf{X}_0, \mathbf{y}_0\right)$$

$$\leq \frac{n}{4} - \frac{1}{4}(\mathbf{y}_1 - \mathbf{y}_0)^\top \psi(\mathbf{z}) + \frac{217}{2304}\|\psi(\mathbf{z})\|_2^2 + \frac{n}{(1+e)^2}$$

$$= c_1 n - \frac{1}{4}\sum_l \psi\left((\mathbf{y}_{1l} - \mathbf{y}_{0l})(g(I-\tilde{\mathbf{A}})(\mathbf{X}_1 - \mathbf{X}_0)_l)\right), \tag{29}$$

where $c_1$ is a constant. For any $\eta$, we construct $\tilde{\eta}_i = \psi_{\frac{1}{g(1-\lambda_i)\delta_i}}(\eta_i)$ such that $|\tilde{\eta}_i g\left(1 - \lambda_i\right)\delta_i| \leq 1$ and $\sum_{i \in \mathcal{I}_{g,\delta,\eta}} \psi\left(\eta_i g\left(1 - \lambda_i\right)\delta_i\right) = \sum_{i \in \mathcal{I}_{g,\delta,\delta,\tilde{\eta}}} \tilde{\eta}_i g\left(1 - \lambda_i\right)\delta_i$. We define $m_g \triangleq \min_{i \in \mathcal{I}_{g,\delta,\eta}} \tilde{\eta}_i \delta_i$.

From the proof of Proposition 1, for any $g(\cdot)$ and $\delta$, we have

$$
\begin{aligned}
\sum_{i=0}^{n-1} \psi\left(\eta_i g\left(1-\lambda_i\right) \delta_i\right) &= \sum_{i \in \mathcal{I}_{g,\delta,\eta}} \psi\left(\eta_i g\left(1-\lambda_i\right) \delta_i\right) \\
&= \sum_{i \in \mathcal{I}_{g,\delta,\tilde{\eta}}} \tilde{\eta}_i g\left(1-\lambda_i\right) \delta_i \geq m_g \sum_{i \in \mathcal{I}_{g,\delta,\tilde{\eta}}} g\left(1-\lambda_i\right) \\
&= m_g\left(\sum_{i \in \mathcal{I}_{\delta,\tilde{\eta}}} g\left(1-\lambda_i\right) + \sum_{i \in \mathcal{I}_g} g\left(1-\lambda_i\right) - \sum_{i=0}^{n-1} g\left(1-\lambda_i\right)\right) \\
&= m_g \sum_{i \in \mathcal{I}_{\delta,\tilde{\eta}}} g\left(1-\lambda_i\right) \\
&\geq m_g \frac{\sum_{i \in \mathcal{I}_{\delta,\tilde{\eta}}} \log\left|\hat{h}_i\right|}{\log \sum_{i \in \mathcal{I}_{\delta,\tilde{\eta}}} g\left(1-\lambda_i\right)\left|\hat{h}_i\right| - \log \sum_{i \in \mathcal{I}_{\delta,\tilde{\eta}}} g\left(1-\lambda_i\right)}.
\end{aligned}
\tag{30}
$$

According to Eq. 29 and Eq. 30, we have

$$
\begin{aligned}
\overline{Er}(\mathbf{X}, \mathbf{Y}) &= \frac{1}{n}\|\sigma(g(I-\tilde{\mathbf{A}})\mathbf{X}) - \mathbf{Y}\|_F^2 \\
&= \frac{2}{n} Er\left(\mathbf{X}_0, \mathbf{y}_0\right) \leq c_1 - \frac{\min_{i \in \mathcal{I}_{g,\delta,\eta}} \psi_{\frac{1}{g(1-\lambda_i)\delta_i}}\left(\eta_i\right) \cdot \delta_i \sum_{i \in \mathcal{I}_{\delta,\tilde{\eta}}} \log\left|\hat{h}_i\right|}{2n \log \sum_{i \in \mathcal{I}_{\delta,\tilde{\eta}}} g\left(1-\lambda_i\right)\left|\hat{h}_i\right| - 2n \log \sum_{i \in \mathcal{I}_{\delta,\tilde{\eta}}} g\left(1-\lambda_i\right)}.
\end{aligned}
\tag{31}
$$

Hence, it completes the proof. □

## D  Algorithm

---
**Algorithm 1** HEROFILTER Framework
---
**Require:** source graph $\mathcal{G}(\mathcal{V}, \mathcal{E}, \mathbf{X})$; adjacency matrix $\mathbf{A}$; patch size $p$; layer number $m$
**Ensure:** $\mathcal{Y}$
  1: **Preprocessing:**
  2: $\tilde{\mathbf{A}} \leftarrow \text{Normalization}(\mathbf{A})$
  3: $\Lambda, \mathbf{U} \leftarrow \text{EigenDecomposition}(\tilde{\mathbf{A}})$
  4: **HEROFILTER Patcher $\phi$:**
  5: $\mathbf{R} \leftarrow \mathbf{U} g(\Lambda) \mathbf{U}^T$             ▷ Compute the patcher score matrix
  6: $\mathbf{P} \leftarrow \mathbf{X}[\text{top-}p_{col}(\mathbf{R}, p)]$      ▷ Extract top-$p$ nodes columnwise from patcher score matrix
  7: $\mathbf{P}^0 \leftarrow \mathbf{P}$                             ▷ $\mathbf{P} \in \mathbb{R}^{n \times p \times d}$
  8: **HEROFILTER Mixer:**
  9: **for** $l \leftarrow 1$ to $m$ **do**
 10:     $\hat{\mathbf{P}}^l \leftarrow \sigma(\text{MLP}(\text{LayerNorm}(\mathbf{P}^{l-1}), \boldsymbol{\theta}_{\text{patch}}^i))$         ▷ Mixing along patch dimension
 11:     $\tilde{\mathbf{P}}^l \leftarrow \sigma(\text{MLP}(\text{LayerNorm}(\hat{\mathbf{P}}^l), \boldsymbol{\theta}_{\text{feature}}^l))$      ▷ Mixing along feature dimension
 12: $\bar{\bar{\mathbf{P}}} \leftarrow \text{Aggregate}(\mathbf{P}^m)$
 13: $\mathcal{Y} \leftarrow \text{MLP}(\bar{\mathbf{P}}, \boldsymbol{\theta}_{\text{predict}})$ ▷ Aggregate along patch dimension via summation function and predict node label.
 14: **return** $\mathcal{Y}$
---

In the standard HEROFILTER framework, we generate patches denoted by $\mathbf{P} = \{\mathbf{P}_v : v \in \mathcal{V}\}$ for all nodes using the patcher $\phi$. The patches $\mathbf{P}$ undergo multiple layers of HEROFILTER Mixer operations to yield $\mathbf{P}^m$, which is subsequently employed for node label prediction to obtain $\mathcal{Y}$. Initially, we preprocess the data by normalizing matrix $\mathbf{A}$, followed by the extraction of $\Lambda$ and $\mathbf{U}$ from the modified adjacency matrix $\tilde{\mathbf{A}}$. Applying the patcher $\phi$ on $\mathbf{X}$ to our graph, we obtain patches of size $[n \times p \times d]$, where $n$ denotes the number of nodes, $p$ the patch size, and $d$ the dimension of the input node features. Each patch, represented by a $[p \times d]$ matrix, encapsulates the extracted features from a node and its top $p$ neighboring nodes. These patches are then processed by

a patch-mixing layer, which mixes information along the patch dimension $p$ by transposing its feature dimension with its patch dimension and passing it through an MLP layer. This process is followed by Layer Normalization and an arbitrary activation function, after which another transpose operation is conducted between the feature and patch dimensions, resulting in a tensor of size $[n \times p \times d]$. This tensor is further processed by a feature-mixing layer, utilizing an MLP to operate on the $d$-dimension. By passing $\mathbf{P}$ through $m$ layers, the final $\mathbf{P}^m$ is constructed. Finally, we aggregate along the patch dimension $p$ of $\mathbf{P}^m$ using an arbitrary aggregation function, yielding a final representation of size $[n \times d]$. An additional MLP is applied to this final representation for classifying the label of each node, resulting in a matrix of size $[n \times |\mathcal{Y}|]$.

---

**Algorithm 2** Fast-HEROFILTER Framework

---

**Require:** Source graph $\mathcal{G}(\mathcal{V}, \mathcal{E}, \mathbf{X})$; adjacency matrix $\mathbf{A}$; patch size $p$; layer number $m$; dangling scalar $c$
**Ensure:** Predicted labels $\mathcal{Y}$
 1: **Preprocessing:**
 2: $\tilde{\mathbf{A}} \leftarrow \text{Normalization}(\mathbf{A})$
 3: **HEROFILTER Patcher** $\phi_{\text{fast}}$:
 4: **for** $v$ in $|\mathcal{V}|$ **do**
 5: $\quad \mathbf{r}_v \leftarrow \text{Minimize } \mathcal{J}(\mathbf{r}_v) \text{ given } \mathbf{A}, c$ $\qquad\qquad\qquad\qquad\qquad\qquad$ ▷ Eq. 12
 6: $\mathbf{R} \leftarrow \text{Stack}(\mathbf{r}_v)$
 7: $\mathbf{P} \leftarrow \mathbf{X}[\text{top-}p_{\text{col}}(\mathbf{R}, p)]$ $\qquad\qquad\qquad\qquad$ ▷ Extract top-$p$ nodes column-wise
 8: $\mathbf{P}^0 \leftarrow \mathbf{P}$ $\qquad\qquad\qquad\qquad\qquad\qquad\qquad\qquad\qquad\qquad$ ▷ $\mathbf{P} \in \mathbb{R}^{n \times p \times d}$
 9: **HEROFILTER Mixer**:
10: **for** $l = 1$ to $m$ **do**
11: $\quad \hat{\mathbf{P}}^l \leftarrow \sigma(\text{MLP}(\text{LayerNorm}(\mathbf{P}^{l-1}), \boldsymbol{\theta}_{\text{patch}}^l))$ $\qquad\qquad$ ▷ Mixing patches
12: $\quad \tilde{\mathbf{P}}^l \leftarrow \sigma(\text{MLP}(\text{LayerNorm}(\hat{\mathbf{P}}^l), \boldsymbol{\theta}_{\text{feature}}^l))$ $\qquad\qquad$ ▷ Mixing features
13: $\bar{\mathbf{P}} \leftarrow \text{Aggregate}(\tilde{\mathbf{P}}^m)$
14: $\mathcal{Y} \leftarrow \text{MLP}(\bar{\mathbf{P}}, \boldsymbol{\theta}_{\text{predict}})$ $\qquad\qquad\qquad\qquad\qquad\qquad$ ▷ Predict node label
15: **return** $\mathcal{Y}$

---

In the Fast-HEROFILTER framework, we generate patches $\mathbf{P} = \{\mathbf{P}_v : v \in \mathcal{V}\}$ for all nodes using patcher $\phi_{\text{fast}}$, mix $\mathbf{P}$ using multiple layers of patch-mixing operations to obtain $\mathbf{P}^m$, and predict node labels $\mathcal{Y}$ using $\bar{\mathbf{P}}^m$. The patcher process involves minimizing an objective function, as described in Eq. 12, to obtain the patcher scores $\mathbf{r}_v$ for each node $v$. These scores are then stacked to form the patcher score matrix $\mathbf{R}$. The dangling scalar $c$ controls how the patcher adapts to different degrees of heterophily during score computation. When $c$ is lower, the focus of the patcher score matrix $\mathbf{R}$ focuses on relationships between nodes with similar spectral properties. On the other hand, when $c$ is higher, the patcher score matrix $\mathbf{R}$ captures relationships across nodes with varying spectral characteristics, allowing the model to adapt to more diverse heterophily patterns. The choice of $c$ enables the framework to flexibly adjust its spectral response based on the underlying heterophily structure of different graphs. The rest of the framework remains the same as the standard HEROFILTER framework.

## E  Experiment Setup

**Data Splits.** In our experiments, we are strictly using the same training and testing environment across different baselines. Following the official data split provided by [45], heterophilic graph datasets (Texas, Squirrel, Chameleon, Cornell, Wisconsin, Actor) and homophilic graph datasets (Cora, CiteSeer, PubMed, OGBN-Arxiv) are split roughly by $48 : 32 : 20$. Two large-scale heterophilic graph datasets (arxiv-year and snap-patents) are following [46], using a $50 : 25 : 25$ random data split. All datasets are provided with masks in the newest version of Pytorch Geometrics.

**Implementation Details.** We employ the Adam optimizer [47] with a learning rate of 0.01 and a weight decay of 5e-4 for training the model. The default training setting for all models is performed using torch-geometric default masks (train, validation, and test) and trained for a maximum of 500 epochs, a hidden dimension of 64, a dropout rate of 0.5, and a number of layers of 2. Early stopping is applied with a patience of 50 epochs, which monitors the validation loss and terminates the training if

Table 8: Dataset Statistics of homophilic and heterophilic graph datasets. The columns show the number of nodes ($|\mathcal{V}|$), the number of edges ($|\mathcal{E}|$), the number of unique classes ($|\mathcal{Y}|$), the feature dimension ($|\mathbf{X}|$), and the average heterophily score ($\frac{\sum h_i}{n}$) indicating the degree of heterophily within each dataset.

| | Dataset | $|\mathcal{V}|$ | $|\mathcal{E}|$ | $|\mathcal{Y}|$ | $|\mathbf{X}|$ | $\frac{\sum h_i}{n}$ |
|---|---|---|---|---|---|---|
| $\frac{\sum h_i}{n} \leq 0.5$ | Cora | 2708 | 5278 | 7 | 1433 | 0.19 |
| | Citeseer | 3327 | 4676 | 6 | 3703 | 0.26 |
| | Pubmed | 19717 | 44327 | 3 | 500 | 0.2 |
| | OGBN-Arxiv | 169343 | 1116243 | 40 | 400 | 0.34 |
| $\frac{\sum h_i}{n} > 0.5$ | Snap-Patents | 2,923,922 | 13,975,788 | 5 | 269 | 0.93 |
| | Arxiv-Year | 169,343 | 1,166,243 | 5 | 128 | 0.78 |
| | Texas | 183 | 295 | 5 | 1703 | 0.89 |
| | Squirrel | 5201 | 198493 | 5 | 2089 | 0.78 |
| | Chameleon | 2277 | 31421 | 5 | 2325 | 0.77 |
| | Cornell | 183 | 295 | 5 | 1703 | 0.89 |
| | Wisconsin | 251 | 499 | 5 | 1703 | 0.84 |
| | Actor | 7600 | 33544 | 5 | 931 | 0.76 |

there is no improvement observed within the specified patience. We employ each baseline method as a representation learner, subsequently concatenating their outputs with a two-layer MLP to perform the node classification task. To assess the performance of the models, we use the standard accuracy metric, which is commonly adopted in node classification tasks.

**Datasets.** We evaluate our model using several standard graph datasets originated from [48, 30, 28, 46, 49, 50]: Cora, Citeseer, PubMed, OGB-Arxiv, Snap-Patents, Arxiv-Year, Texas, Squirrel, Chameleon, Cornell, Wisconsin, and Actor. These datasets are diverse in terms of their sizes and degrees of heterophily. Specifically, homophilic graph datasets are Cora, Citeseer, and PubMed, while heterophilic graph datasets include Texas, Squirrel, Chameleon, Cornell, Wisconsin, and Actor. To demonstrate the effectiveness of HEROFILTER on larger scale graph datasets, we also evaluate our model on Penn94 [46], Arxiv-Year, OGBN-Arxiv [49], and patent networks [50] datasets.

**Comparison Methods.** We benchmark our model against a variety of established methods. For homophilic graphs, we consider: GCN [19], GAT [20], GPRGNN [37], ChebNet [21]/ChebNetII [35], APPNP [31], GCN-JKNet [51], GraphSage [52], and FAGCN [23]. For heterophilic graphs, we explore: $H_2$GCN [32], BM-GCN [33], BernNet [44], and $G^2$-GCN. For graph transformer based methods, we compare with: NAGphormer [38], VCR-Graphormer [41], Exphormer [40], Polyphormer [42] and GOAT [39]. Lastly, we incorporate MLP.

## F  Details of Parameter Settings and Tuning

We perform a grid search to find the optimal hyperparameters for our model, including the learning rate, patch size, and filter number. The search space is defined as follows: the learning rate $\in \{0.01, 0.008, 0.005, 0.003, 0.001\}$, the patch size $\in \{8, 16, 32, 64, 96\}$, and the filter number $\in \{10, 50, 100, 150, 200\}$. For Fast-HEROFILTER, we set $c$ as 0.5 to balance the spectral response across different frequency components. The optimal hyperparameters are determined based on the performance of the validation set.

## G  Scalability and Runtime Analysis

**Scalability Analysis on HEROFILTER Mixer.** We analyze the runtime of the algorithm as the number of nodes and patch size varies to assess the scalability of our proposed approach in Figure 4a. We perform experiments on synthetic datasets with varying numbers of nodes from {10, 100, 1,000, 10,000, 100,000}. We also consider different patch sizes, specifically {8, 16, 32, 64}. The runtime results are reshaped into a matrix for visualization purposes. Figure 4a shows the runtime in seconds on a logarithmic scale for the different patch sizes as a function of the number of nodes. As expected, the runtime increases with the number of nodes and patch size. Nevertheless, Fast-HEROFILTER demonstrates reasonable scalability as the growth in runtime is sub-linear, indicating that our approach

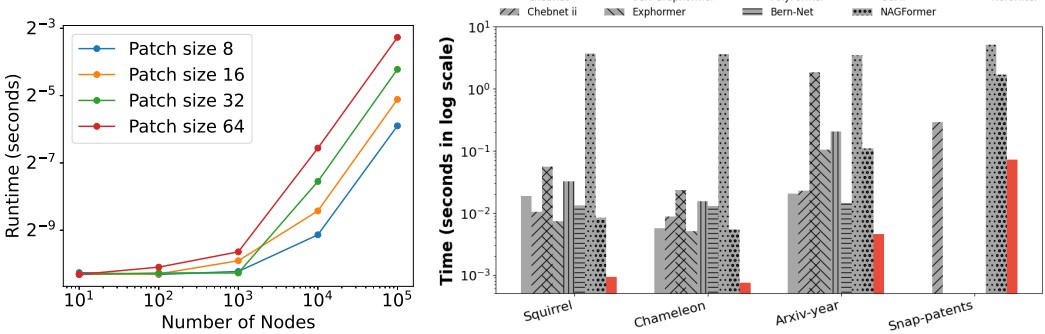

(a) Scalability analysis of Fast-HEROFILTER. The x-axis shows the number of nodes processed, and the y-axis shows the running time per iteration.

(b) Runtime results for one epoch of inference (5 runs). Lower is better.

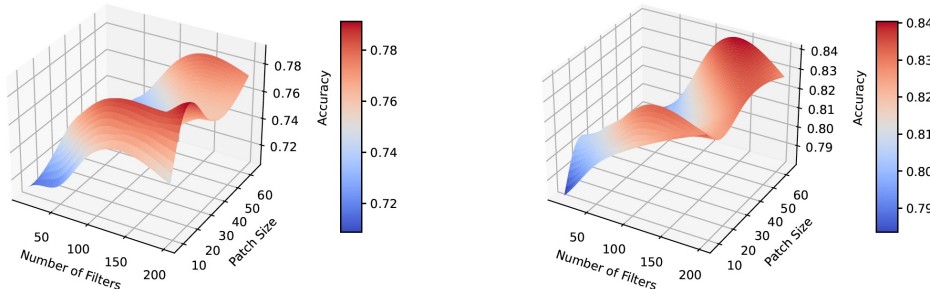

Figure 5: Parameter sensitivity analysis results for Cora (left) and Texas (right) datasets, respectively.

can handle large-scale graph datasets effectively. The results also highlight the importance of selecting an appropriate patch size to balance the trade-off between computational efficiency and performance.

**Runtime Analysis on HEROFILTER Mixer.** We further evaluate the runtime performance of HEROFILTER against existing filter-based and graph transformer methods across four datasets: Squirrel, Chameleon, Arxiv-year, and Snap-patents. Since HEROFILTER patcher can be computed offline as a preprocessing step, we only record the inference time of HEROFILTER mixer. Shown from Figure 4b, our findings reveal that HEROFILTER consistently achieves the fastest inference time across all datasets. Specifically, on Squirrel and Chameleon, HEROFILTER executes in 0.92 and 0.75 milliseconds, respectively, outperforming the next fastest method (Exphormer) by 8 times. For larger datasets like Arxiv-year and Snap-patents, HEROFILTER maintains its efficiency (45.60 and 71.66 milliseconds), while transformer-based methods like VCR-Graphormer either require significantly more time (1832.73 milliseconds) or fail to complete due to out-of-memory (OOM) errors.

# H   Parameter Sensitivity Analysis

We conduct a parameter sensitivity analysis to investigate the impact of patch size and the number of filters in Figure 5. The analysis was performed on two datasets, namely the Cora and Texas datasets. We varied the patch size from {8, 16, 32, 64} and the number of filters from {10, 50, 100, 200}. We measure the accuracy of the model for each combination of parameters. Figure 5 shows the results of the parameter sensitivity analysis. The plots reveal that for both datasets, increasing the patch size and the number of filters generally leads to higher accuracy, although the improvements tend to plateau beyond certain values. This indicates that our model can effectively handle different patch sizes and filter configurations while highlighting the importance of selecting appropriate parameters to achieve optimal performance.

